# Gardens of Historic Mental Health Hospitals and Their Potential Use for Green Therapy Purposes

**Anna Staniewska** 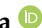

Chair of Landscape Architecture, Faculty of Architecture, Cracow University of Technology, Warszawska 24, 31-155 Kraków, Poland; astaniewska@pk.edu.pl

**Abstract:** Gardens of historic psychiatric institutions represent a special type of heritage garden that possess both aesthetic and therapeutic purposes. Their existence and current state are affected by changes in the organisation of mental treatment. The article focuses on the possible use of these gardens as places of modern green therapies carried out in, and connected with, nature. Taking into account the state of the art on the beneficial influence of nature on human health and well-being, the paper provides an overview of historic and modern nature-based activities considered therapeutic. Subsequently, three case studies of contemporary psychiatric facilities operating in historic mental hospital sites are examined. Many activities linked to nature exercised historically in those gardens bear similarities to a contemporary spectrum of ecotherapies. An analysis of historic and contemporary plans of the sites and gardens and a description of the therapeutic activities carried out in nature are provided. Results prove that their potential is promising, but not yet used to its full extent because of organisation and financing within the context of health care systems. Using those gardens for the spectrum of green therapies may bring benefits for patients and the historic substance alike.

**Keywords:** historic psychiatric hospitals; therapeutic landscape; green therapy; horticulture therapy





## 1. Introduction

### 1.1. Historic Psychiatric Hospitals and Their Gardens—From Lunatic Asylum to Contemporary Mental Health Centres

Historic psychiatric hospitals were built in the early 19th century as specific places of treatment of manifold disorders described usually as 'lunacy', during the era of psychiatry's beginnings as a distinct medical specialization. Lunatic asylums—as they were called—represented special types of buildings and landscapes, which were designed according to the theoretical works of medical doctors and took into account their experiences gathered while managing those institutions and collected during study travels undertaken to visit model institutions. With time, these institutions evolved just like psychiatry developed. At the beginning of the 19th century, a significant advance was brought by the writings and practice of Pinel in France and Chiarugi in Italy [1,2], and the introduction of moral treatment practised in England in a private Quaker institution called The Retreat and led by William Tuke, who handed it to his grandson Samuel, who contributed greatly to publicizing his achievements [3]. The humane attitude implemented at this small establishment was widely propagated and treated as a model for many following decades. It was based on the assumptions that the asylum's interior and surroundings should be pleasant to distract the troubled mind, and that appropriate daily routine, diet, and occupation contributed to healing.

The structure of the asylum and its landscape was similar to an English country house estate [4], but it had a clear therapeutic function. Although asylums evolved, some principles remained the same. Gardens of historic psychiatric institutions represent a special type of historic garden that served not only aesthetic but also therapeutic purposes [5,6]. At

the turn of the 19th and 20th centuries, psychiatric hospitals were aspiring to be more than just institutions of isolation, aiming at providing a truly restorative setting with countryside locations, and accommodating patients in detached pavilions instead of large corridor buildings. In addition, occupational therapy was widely introduced along with agricultural and horticultural activities on hospital farms. Various types of physical activities, such as *'tennis, bowls, badminton, football and amusements'* (such as the three-legged race) were also considered beneficial and advertised in theoretical writings [7] (pp. 100–114).

Interestingly, radial plans of several mental institutions from the beginning of the 20th century in Europe visually resemble a diagram of garden cities as presented in the writings of Ebenezer Howard (Figure 1). While no direct inspiration for the design and building process is usually found in historic sources [8], this type of circular plan and similarities to garden cities contributed to these institutions being described as 'garden cities for the insane' [9].

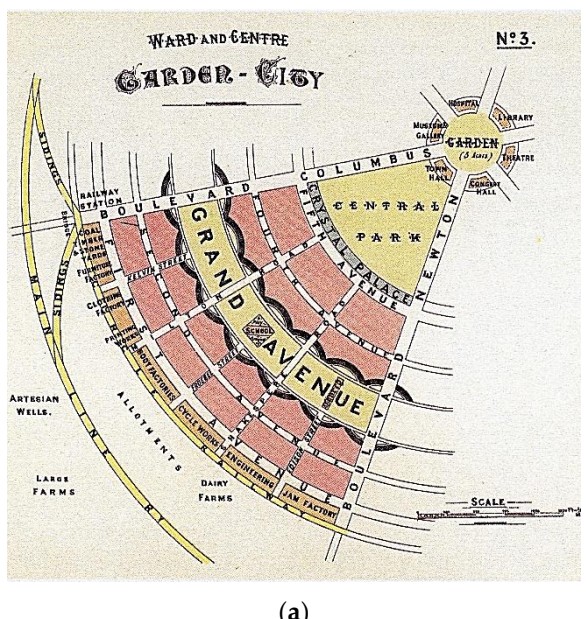

(**a**)

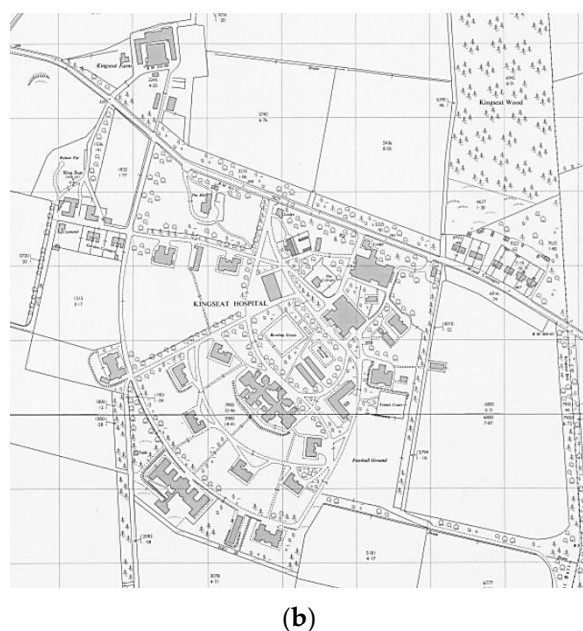

(**b**)

**Figure 1.** Similarities in plans between the garden city diagram and the plan of a mental asylum of the early 20th century: (**a**) Diagram No. 3 by Ebenezer Howard showing the ward and centre of the garden city (from Howard, Ebenezer, *To-morrow: A Peaceful Path to Real Reform*, London: Swan Sonnenschein & Co., Ltd., 1898; source: [10] public domain, (**b**) The plan of the Kingseat Asylum in Scotland, built from 1900 to 1904, as pictured on the Ordnance Survey Map from 1944; source: the collection of the National Library of Scotland, CC-BY (NLS).

Despite all these efforts and the therapeutic regime implemented by early psychiatrists working there, asylums were heavily criticized by the advocates of clinical treatment, which was considered more effective than alienists' work. Finally, large institutions were defeated by numbers, because in an institution that sometimes holds as many as 2,000 patients, any therapy was condemned to failure [11]. The first three decades of the 20th century brought an almost double increase in the number of psychiatric patients in Britain [12]. Despite the introduction of brain surgery and electric shock treatment, psychiatric patients demonstrating socially unacceptable behaviour were administered sedatives, and the era of confinement in large psychiatric institutions lasted until the 1950s. A significant breakthrough in the treatment of mental disorders was achieved with the humanitarian effect of antipsychotic drugs. This, combined with a strong antipsychiatry movement [13], contributed to the process of change and closure of the asylums.

In the UK, the decision to close Victorian asylums was strongly linked with the famous 'Water Tower speech' delivered by the minister of health, Enoch Powell, at the annual conference of the National Association for Mental Health in March 1961. Referring

to specific elements of asylum sites, he described these institutions as *'isolated, majestic, imperious, brooded over by the gigantic water-tower and chimney combined, rising unmistakable and daunting out of the countryside'* [14]. Services provided by psychiatric hospitals were to be delivered by special acute-care psychiatric wards of general hospitals and community care. This resulted in the dissolution of the asylums and plans to reduce the number of mental health beds. This policy, in terms of patient comfort and quality of mental healthcare services, has brought mixed results [15]. While, generally, large-scale institutions were heavily criticized for their overcrowding and scandalous cases of violent treatment of patients, and community care had its evident benefits, the fate of the elderly and patients requiring long-term treatment, who were permanent residents of old institutions, became difficult [16]. This wave of asylums' closures was called deinstitutionalization, and led, in many cases, to the complete demolition or redevelopment of numerous Victorian asylum sites. Many of them stayed vacant for a long time, because it was difficult to obtain planning permission for redevelopment, which significantly increased the value of the area [17,18]. Many were irreversibly destroyed and demolished, while numerous sites were turned into up-market housing developments where all traces of the old asylum were often deliberately obliterated [19,20]

Deinstitutionalization of psychiatric hospitals affected not only UK institutions, but also historic asylums located in Italy, the United States, Australia, and New Zealand [21,22].

Redevelopment strategies for former asylum sites' conversion usually focus on their attractive locations (on the edge of the city, with pleasant views, and greenery in abundance) and they often lead to the process of strategic forgetting of the past, or to selectively remembering it to repackage the asylum as housing [23].

Yet, some contemporary mental health centres still operate in historic psychiatric hospitals in several European countries, such as France [24], Switzerland [25], Germany [26], and Poland [27]. They offer the most up-to-date treatment in the renovated buildings surrounded by historic green spaces and parks.

The existence and current state of historic mental hospital gardens is inevitably bound to changes in the organisation of mental therapies and treatment. What is more, like many other historic gardens, they are nowadays facing numerous challenges linked with the need for proper maintenance and issues connected to climate change, along with the ageing of the original planting. The potential use of their gardens for the spectrum of green therapies seems to bring benefits for patients and the historic substance alike, although it is not fully used.

### 1.2. Aim of the Study

The article presents the therapeutic potential of historical gardens and parks of former psychiatric hospitals, indicating the possibilities of using these gardens to conduct green therapies. Furthermore, it explores some problems related to the maintenance of historical vegetation, focusing primarily on the main benefits of nature-based interventions.

Links between nature and health have been explored since ancient times dating back to Hippocrates, who in *On Airs, Waters, and Places* referred to *'the body [which] was inseparable from inquiry into places and directions, seasons and winds [ . . . ] human being was being embedded in a world'* [28] (p.661). Early research was based on several theories that refer to medical geography. The most important of them focus on the concept of therapeutical landscapes [29,30] and evolutionary perspectives that encompass the prospect-refuge theory [31], the functions-evolutionary model [32,33], and the psychoevolutionary model of R. Ulrich [34–36].

Recently, numerous studies have explored not only the historical and modern paradigms (of the various disciplines) which determine the discourse of nature concerning human health and well-being research [37], but have referred to a multitude of particular aspects connected with these issues. While gardens have always been an appreciated element [38] of historic cities, providing public green leisure areas such as commons, pleasure grounds, and public parks [39], they have nowadays become a vital part of green infrastructure

(GI) [40] which may provide, among other services, therapeutic functions. Historic gardens, along with other types of garden spaces [41,42], may be treated as resources in sustainable urban development in a time of accelerating densification and climate change. Relationships between green infrastructure and ecosystem, and human health, construction, evaluation and management of green infrastructure, and analysis of a special aspect of green infrastructure are research topics in the field of landscape planning and urban studies that are of great importance recently [43]. Many studies explore the benefits of nature [44–46] and sustainable urban living spaces that are vital in an urban context, as the population of urban dwellers grows on a global scale and is confronted with the climate crisis [47,48]

Apart from the studies focusing on physical health [49], there is a growing body of research that focuses on a positive association between urban green space and attention, mood, and physical activity [50]. Other research aims at measuring the impact on mental health of relations to, and activities carried out in, natural environments [51]. They often examine the link between visiting green spaces and well-being [52], indicating that more time spent in green spaces is associated with better mood [53] and higher scores on mental health and vitality scales, independent of cultural and climatic contexts [54]. An important role is especially attributed to physical activity during leisure in green space [55], and interactions with nature, such as walking [56], which are likely to bring numerous benefits, especially for people with mental issues such as depression [53].

Research on various nature-based interventions (NBIs) is of vital importance, as mental disorders are increasingly diagnosed in recent years. For this reason, therapeutic measures connected with nature are gaining importance. The WHO states that in 2019, one in every eight people, which equals nearly 970 million people around the world, were living with a mental disorder [57], and estimates that the COVID-19 pandemic contributed to an increase in anxiety and major depressive disorders in only one year [58]. In the United States, 19.86% of adults are experiencing a mental illness, which is equivalent to nearly 50 million Americans [59].

Because natural healing powers are particularly relevant in the case of stress reduction, a wide spectrum of NBIs seem to present promising opportunities to support not only treatment, but also the prevention of several mental disorders. Various stress-related disorders [60] may be caused by psychosocial stress, such as fatigue, burnout, exhaustion, depression, anxiety, or adjustment disorder [61].

While most of the research referring to horticultural therapy was carried out to examine the impact of gardening on vulnerable groups, such as the elderly [62], children and youth [63], patients with mental illness [64] or particular problems such as depression and related issues [65], PTSD [66–69], and dementia [70,71], only some attention was paid to the setting in which the therapy was conducted [72]. Of course, the features that define a therapeutic garden [73–75] and park [76] are formulated, but they are applied mostly to newly established environments.

There are no studies on therapeutic activities supporting psychiatric treatment carried out in the parks and gardens of historic psychiatric hospitals, and relations between their heritage values and composition. Therefore, this study aims to describe relations between the landscape and garden composition of heritage mental hospital sites and a spectrum of green therapies in the past and nowadays. Green-based interventions in those particular places will be examined with reference to historic sources, the contemporary literature, and research. In addition, three selected case studies of contemporary mental health centres operating on historic psychiatric hospital heritage sites will be examined.

## 2. Materials and Methods

To explore the links between the historic mental hospital landscapes and contemporary green therapies supporting mental health and well-being, the study employs a mixed-methods approach and is based on secondary desk research and structured interviews, which together contribute to case study analyses and provide data for discussion.

Firstly, the connections to nature offered in the past, in former asylums from the beginning of the 20th century, are explored. This part of the paper is based on historic materials and included primary sources, such as writings of medical doctors and asylum superintendents summarised previously in secondary sources, including the previous body of research on the landscape composition and arrangement of historic asylums and their gardens in Europe [6,77].

Subsequently, the contemporary spectrum of green therapies widely described in recent reviews is analysed with reference to the historic use of green spaces and activities carried out in historic psychiatric hospitals, and possible implementation within the historic model.

The last stage is an exploratory case study based on structured interviews carried out in three mental health facilities which operate in historic buildings located in a park setting. It describes the location and brief history of each particular hospital and provides information on the contemporary spectrum of green therapies conducted in those selected sites. Structured in-depth interviews were used to collect relevant source data to obtain comparable materials and qualitative information on the types of green therapies conducted and their relation to the heritage park site. It was particularly important if nowadays historic landscape features were intentionally used. Interviews are an acknowledged method of data collection in many research fields [78–80] including heritage science [81] and medicine alike [82]. As an interactive method, an interview enables mutual learning and allows the researcher to focus on the issues that the interviewer might not have previously considered important [83].

In the discussion, reference is made to other relevant examples of historic hospital settings and their therapeutic use, although these may not exactly match the case study criteria.

*Case Study Choice and Location*

Although numerous contemporary institutions still operate in historic buildings and sites of old psychiatric hospitals across Europe, three were chosen for this study. The case studies were selected to present the most developed places where all possible elements of the functional programme were implemented at the stage of the design. For this reason, mental hospital sites which were built in 1900 or later were chosen. In addition to this, selected hospitals represent institutions built in pavilion style on a circumferential plan resembling the garden city diagram—an example of harmonious development which resembles a small community—and which still operate nowadays. In addition, they are of comparable size in terms of surface. The original plans of those establishments were well preserved and accessible. All three institutions declare that they appreciate their historic setting and try to make contemporary use of historic green spaces, and that they adapt heritage buildings to current treatment and patients' needs. The following mental health institutions (Table 1) were selected for investigation of the contemporary green therapies actions in a historic setting:

1. The Babiński Specialized Hospital in the southern suburb of Kraków, former second National Institute for the Nervous and Mentally Ill, Kraków-Kobierzyn (Małopolska Voivodeship, Poland);
2. The Voivodeship Mental Hospital Lubiąż, former Provincial Hospital for the Nervously and Mentally Ill (so-called 'New Institution', to differentiate it from the old hospital situated in a former Cistercian abbey), Lubiąż (Lower Silesia Voivideship, Poland);
3. Klinikum am Weissenhof Centre for Psychiatry Weinsberg, former Königliche Heilanstalt Weinsberg, Staatliche Irrenanstalt, Weissenhof-Weinsberg (Baden-Würtemberg, Germany).

**Table 1.** An overview of the historic mental hospitals and their gardens under study.

|  | Kraków-Kobierzyn | Lubiąż | Weissenhof-Weinsberg |
|---|---|---|---|
| Historic name | The second National Institute for the Nervous and Mentally Ill | Provincial treatment and care facility in the town of Lubiąż | Königliche Heilanstalt Weinsberg, Staatliche Irrenanstalt |
| Contemporary surface in ha | 48 ha | 21 ha | 45 ha |
| Construction dates | 1909-1919 | 1902-1910 | 1900-1903 |
| Architects involved | Władysław Klimczak | Eduard Blümner | Carl Hees |
| Landscape architects | Wiktor Żochowski | nn | Albrecht Lilienfein und Sohn |

## 3. Results

Activities in nature have been used as means of therapy in the past, and they were taken into account as important features to be considered when establishing an asylum from the turn of the 19th and 20th centuries.

This section provides an overview of links to nature-based activities in the past and a broad spectrum of contemporary opportunities for green therapies, and places them in the context of historic mental hospital landscapes. Subsequently, three case studies are described with particular attention to their landscape composition and layout, and green therapies conducted nowadays.

### 3.1. An Overview of Connections to Nature and Green Therapies Historically Used in Mental Hospitals from the Turn of the 19th and 20th Centuries

Firstly, the setting of the institution was important: on the outskirts of the city, in the country, and preferably on a hill [84] with beautiful views to ease the troubled mind and isolate the mentally ill from harmful industrialised and overpopulated cities [85]. Gardens often used borrowed views from elevated mounds to secure the pleasant, picturesque views for patients without being seen [86] (pp. 191–192). To restrict the visibility of curious onlookers, the institutions were surrounded by fences and walls often constructed as ha-has [87] (p. 185). Water was also important, and was presented as a fountain or a well in the middle of the rectangular inner court or garden [88] (pp. 114–115).

A range of activities was encouraged and carried out on the hospital grounds. Walking took place in airing courts; however, it was restricted to the wards for agitated and criminal patients. In Brislington House, which was located in the vicinity of the Avon River, therapy included walks in nature [89], and some institutions even organised excursions for patients (such as Hanwell Asylum [84], or Illenau, near Achern, in Germany [88]).

Physical exercise was encouraged in the airing courts and on the hospital grounds (battledore, shuttlecock, football, and cricket). In many places there were special amenities designed within the parks, such as a bowling green and football pitch, and even tennis courts for upper-class patients (as reported by the director of the Kobierzyn institution in the 1920s [90]).

More sophisticated activities were often aimed at and exercised by educated, wealthy, and noble patients. An interesting collection of drawings and paintings by Charles Altamont Conan-Doyle (father of Sir Arthur Conan-Doyle), who spent several years in Montrose Asylum in Scotland, is well known [91].

John Conolly, who introduced occupational therapy while acting as superintendent in Hanwell Asylum, admitted that gardening was one of the meaningful occupations for patients, who took care in turning airing courts into gardens and made representative entrance areas beautiful. He wrote:

*The cultivation of the gardens, and of the ground called the farm, as well as of the extensive ornamental ground in front of the asylum, is entirely effected by the labour of numerous male patients, superintended by gardeners, or by steady workmen. The cheerfulness with which their work is performed, and the satisfaction with which, at stated hours, they assemble for their allowance of beer, sufficiently attest that calming and remedial influences are thus exercised* [84] (p. 51).

Many other asylums employed some of the patients in agricultural work. Obviously, nowadays a strong tension can be observed between the primary therapeutic aim of these activities and the economic necessity to provide the institution with its fruit, vegetables, and dairy products. The numbers of patients involved in the work on hospital farmland or in so-called 'agricultural colonies' varied. As Sarah Rutherford discovered, it was on average up to 30%. But there was an intention that regular work in the open air was not only to keep patients busy with the purposeful work, but also to provide them with visible satisfaction from the results of their work, physical exercise, and fresh air. Interestingly, hospital records cited by Rutherford show that the productivity of hospital farms was higher in the Broadmoor institution, which was a criminal asylum, than in less corrective asylums such as Hanwell [4] (pp. 226–228). While some doctors advising on the theory of insane asylum construction primarily considered an asylum farm as an important element of the institution's functioning, some indicated that animal keeping could also evoke positive feelings and cheerfulness, and mentioned that some animals could be kept for pleasure and entertainment including, among others, *'rabbits, sea-gulls, hawks and poultry'* [3] (p. 63).

All those activities were conducted in separated male and female groups, taking into account social norms and cultural habits attributed to some of them, especially to men or women. Women were employed in all types of housework in the kitchen and vegetable kitchen garden, in addition to lighter work in the fields, gardening, washing, and repairing laundry. Other types of occupational therapy included wickerwork and linen production. Men were involved in heavier agricultural, building, and construction work, often helping to complete the asylum complex, which saved large sums of money, as in the case of Brassens in France [92] (p. 62).

In institutions which also admitted tuberculosis patients, there were often verandas and special sunny terraces or so-called 'Liegehallen' [93] that facilitated passive rest with a view of the garden for those suffering from pulmonary problems.

*3.2. Nature and Green Therapies Supporting Mental Health and Well-Being Nowadays*

Therapeutic activities which use nature and natural materials or natural settings, also referred to as nature-based interventions (NBIs), were often traditionally employed as calming and relaxing activities in historic asylums. In the last 50 years, they fall under two most frequently used umbrella terms: green therapies and ecotherapy. While green therapies and NBIs describe a range of outdoor activities which use plant material, planted garden settings, or natural environments, ecotherapy results from the deeper philosophical concept of exploring particularly strong bonds and mutual relations between humans and nature [94]. Ecotherapy is sometimes referred to as 'a forgotten ecosystem service' [95] and, as such, is considered as offering a range of therapeutic programmes based on NBIs [96], which support conventional treatment by reducing the number of pharmaceuticals [97] and preventing mental health problems.

Ecotherapy is often defined as therapeutic treatment consisting of regular structured activity, led by trained professionals, taking place in a green environment, focused on performing an activity, and relating to exploration and appreciation of nature in its various forms and aspects. It often involves social contacts generated by spending this time with other people, although the interactions are never forced. It includes a few main groups of programmes (see Figure 2), such as:

- Social and therapeutic horticulture (passive: simply spending time and admiring gardens and plants; or active: focused on gardening, tending the food-growing plants. May also take place indoors, in greenhouses);

- Green exercise therapy (doing exercise in green spaces: yoga, walking, running, or cycling);
- Care farming (the therapeutic use of agricultural landscape and farming practices such as growing crops, looking after farm animals, or helping to manage woodland);
- Animal-assisted interventions (spending relaxed time in contact with animals in spaces like farms, especially introduced in groups of young patients);
- Animal-assisted therapy (meant as building a therapeutic relationship with animals, especially dogs and horses);
- Environmental conservation (activities focused on protecting and caring for natural spaces, often combining physical exercise with conservation tasks);
- Nature arts and crafts (creating art in green spaces, or with nature and natural materials, such as painting, sculpture, and creating land-art, and also referring to use of the environment as inspiration);
- Adventure therapy (focused on adventurous physical group activities like rafting or rock climbing);
- Wilderness therapy (spending time in the wild and in remote locations, performing activities together in a group, such as making shelters and hiking) [98,99].

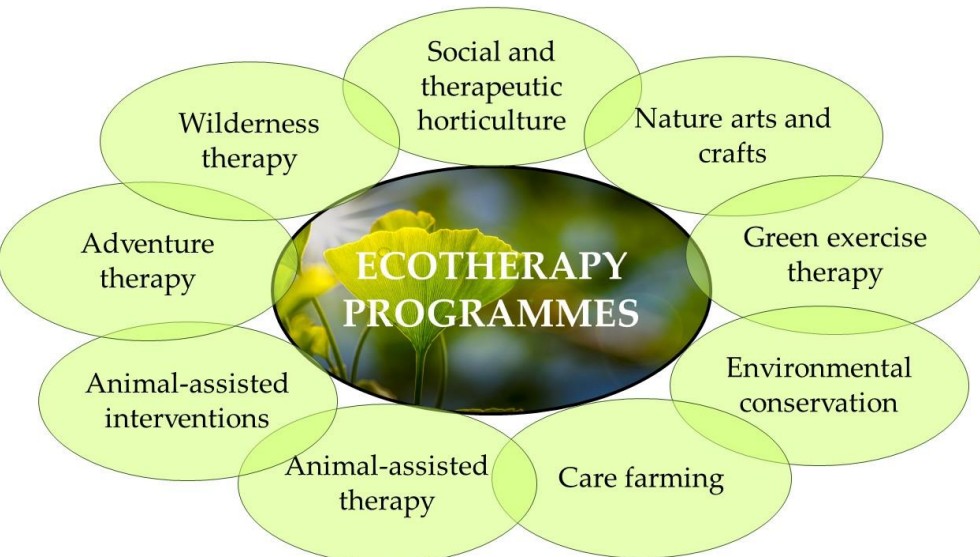

**Figure 2.** A variety of ecotherapy programmes, own elaboration based on a paper by MIND [98,99].

*3.3. A spectrum of Green Therapies Conducted in Selected Contemporary Facilities Operating within the Walls of Historic Psychiatric Hospitals*

3.3.1. Kobierzyn, Krakow

The Babiński Specialized Hospital in Kobierzyn was established as the second National Institute for the Nervous and Mentally Ill in the western part of Galicia, a province of the Austro-Hungarian monarchy at the beginning of the 20th century. In 1903, a provincial diet decided that a suitable plot for building a 500-bed institution for the nervously and mentally ill should be found, and initial plans, along with cost estimates, should be prepared within a year. For several reasons, the process was delayed and a final resolution on building the hospital was passed by the parliament in the of autumn 1907. Doctor Jan Mazurkiewicz, a recognized specialist, was appointed as the director. The plan for the Kobierzyn hospital was initially prepared in Lviv under the supervision of Władysław Klimczak, a professor of the Lviv Polytechnic University, appointed as the construction manager. In April 1910, however, the entire hospital planning office was moved to the site at Kobierzyn, and construction of the pavilions began. The setting on the southern outskirts of Kraków offers distant views to the south, in the direction of Beskidy mountains, and to the north, on the valley of the Vistula, the city, and the woods of the so-called western green-wedge, with the hills of Sikornik topped with Kościuszko Memorial Hill. The hospital buildings and

their gardens occupy more than half of the whole hospital area which covers about 48 ha. The hospital was constructed in the pavilion style and all units were immersed in the vast park. The hospital also had its farm and arable land (on the southern side), and a cemetery in the vicinity (apart from the main hospital area to the north). The hospital was built to accommodate 550 patients, with a possible extension to handle up to 800. Because it was built within the firing range of the southern front of the Krakow Fortress, it was in danger of being demolished in 1914, and it was only in early 1918 that the first patients were admitted [100].

Regarding landscape design, the hospital possesses the plan of the gardens and park from 1909, signed by Wiktor Żochowski, a co-owner of a gardening company operating in Kraków (Figure 3). This design was only partially implemented in a much-simplified form. While the main alleys, very simplified roundabout parterres, and geometric ward gardens are visible on an aerial picture taken in 1917, the rest of the landscaping was done after World War I, as mentioned in reports of the next director in 1925. Most of the trees that form the historic garden layout date from this time. During World War II, the hospital was under national socialist occupation and many patients were exterminated. The facility reopened after the war, and the number of patients increased significantly. The institution experienced some interventions in its main composition and structure, but still operates as a psychiatric hospital. Although agricultural work and land cultivation were carried out for a long time, they were finally abandoned as unnecessary for the continuation of the medical mission. Since 1988, the ensemble has been listed as a registered monument. Over the last 25 years, restoration of buildings has been intensified as necessary to improve conditions and the comfort of patients, and numerous treatment programmes reduced the number of patients requiring long-term stays on the wards [101].

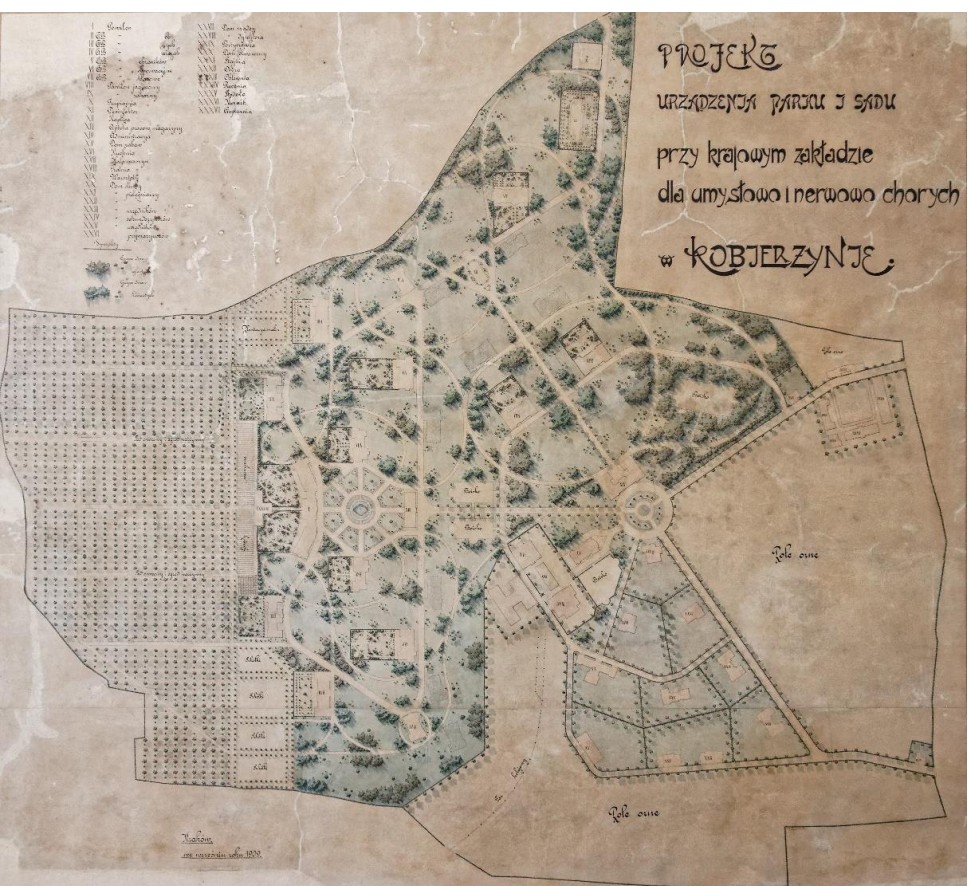

**Figure 3.** A plan of the establishment of a park and orchard around the National Institute for the Nervous and Mentally Ill in Kobierzyn, Kraków, 1909, drawing by Wiktor Żochowski; source: the archive of the Babiński Clinical Hospital.

Therapeutic gardening sessions in Kobierzyn hospital are carried out regularly once or twice a week in each ward, and they usually last one hour. The number of patients taking part in therapeutic gardening varies from 3 to 26 people, depending on the condition of patients in a given ward. Monthly, from 200 to 570 people participate. Since 2014, the number of patients involved in therapeutic gardening grew systematically from 11.2% to 29.5% in 2018 [102].

Within the framework of the garden therapies, two main types of sessions are offered. The first 'active' type is conducted in the ward gardens if the weather allows it. It covers the full range of activities performed to set up and maintain the gardens. If the gardens require revitalization or re-establishment (e.g., after a general renovation of the building), the ground for decorative flower beds is prepared by the patients acting together, and covered with garden cloth. Ornamental plants are planted and bark is laid down to prevent weeds.

Patients also regularly maintain the flowerbeds, undertaking care consisting of systematic weeding, pruning, fertilizing, and preparing beds for winter (covering plants to protect them from frost). Therapeutic gardening is also conducted in a greenhouse, which belongs to the Social Cooperative 'Kobierzyn' (social enterprise). In the greenhouse, patients learn to produce seedlings of vegetables and ornamental plants, and to make cuttings from ornamental shrubs. Garden therapy also includes dendrological walks, which aim to show patients the species of trees and shrubs that grow in the historic hospital park. Patients also collect and dry flower petals, which they use later during creative workshops in winter to create artwork and pictures. On the other hand, in winter and when it rains, hortitherapy is conducted in a 'passive way', via multimedia content on topics connected to the history of gardens and gardening issues. Patients become acquainted with the examples and pictures taken in the most beautiful gardens in Poland and abroad. Presentations also cover the issues of the health properties of various types of fruits and vegetables, along with the wealth of ornamental plant species, as well as birds and animals visiting the gardens. In addition, in winter, the seeds collected in autumn are cleaned and sorted to be used in the next year to create flowerbeds in the gardens of the wards. In addition, the garden therapist runs workshops for patients on designing decorative flower beds and herbal gardens.

Patients appreciate the historic park while taking walks and admiring nature, which helps them to relieve stress and escape from trauma. In the gardens of the wards, they organize barbecues during the warm time of the year. This is also a place to sit during visits by family and friends, in the case of a closed ward patient. Depending on the weather, patients spend several hours a day in these gardens. Most of them are on the southern side of unit buildings and are equipped with benches and tables, while a few have pavilions protecting against sun and rain [103].

The therapists are particularly fond of the gardens adjoining the wards as daily accessible green spaces and are happy to consult the experienced gardener and provide guidance on professional tools and equipment.

The sports ground is used occasionally for the annual sports celebration (meeting). In 2019, open-air gyms with some equipment were installed, and a chess table, which can be used by all visitors to the hospital grounds (two locations).

During the COVID-19 pandemic, some spaces were used. The unit gardens were treated as an extension of the ward and patients used them. During the first lockdown phase (from March to May 2020), no visitors were allowed to the park and patients did not go for any walks. In addition, the therapeutic gardening groups were suspended, but the therapists looked after the gardens, which they appreciated as a beneficial activity helping them to regenerate after work.

### 3.3.2. Lubiąż

The construction of the Provinzial heil- und Pflege-Anstalt Städtel Leubus (provincial treatment and care facility in the town of Lubiąż) was approved by the Parliament of the Province of Silesia in the early spring of 1901. It was supposed to be a new facility

(often later called *'das Neue Anstalt'*) located approximately two kilometres from the facility established in 1830, and operating in a secularized post-Cistercian Baroque monastery complex. The selected plot was located on a low moraine hill on the outskirts of the small town of Lubiąż, near the old crossing on the Odra River, with good access to the main road leading to the town and the monastery. There was, however, one disadvantage of the location: the nearest railway stations were 8, 9, and 15 km away. The author of the project was Eduard Blümner, an architect and building construction counsellor (Baurat) from Wrocław specializing in public facilities (and hospital and care buildings). About 30 buildings were located in the initial area of 151.6 ha, which could serve nearly 1000 lower-class patients as well as employed doctors and staff. Construction began in 1902, the first patients were admitted in 1906, and the institution was completed in 1910 [104]. Now, the surface of the facility is significantly smaller (ca. 21 ha), since arable land was excluded.

The hospital buildings were arranged in a geometric, central-axial layout, and the pavilions for patients built on the peripheral alley were oriented in such a way that the patients' rooms and day rooms had windows opened to the southwest, and the verandas and terraces extended towards the pavilion gardens. The main axis of symmetry ran from the gatehouse at the entrance, then between the doctors' house and the admissions pavilion and the clerks' house, and up to the round lawn with a flower bed, in the centre of which stood the *Festsaalgebäude*—a building that served as a theatre, playroom, and concert hall. Behind the theatre, there was a kitchen and an ice cellar, and the axis was closed by the contagious pavilion, preceded by the symmetrical bipartite garden. The axis also divided the hospital complex into sections for women and men. On the female side, from the southeast, apart from the main part of the complex, there was a boiler room, a laundry room, and another house of doctors, while on the northwest side there was a farm outside with the inspector's house and livestock buildings (stables, pigsties, and cowsheds). Outside the layout, on the eastern side there was a water intake with a reservoir and a cemetery with a chapel (Figure 4).

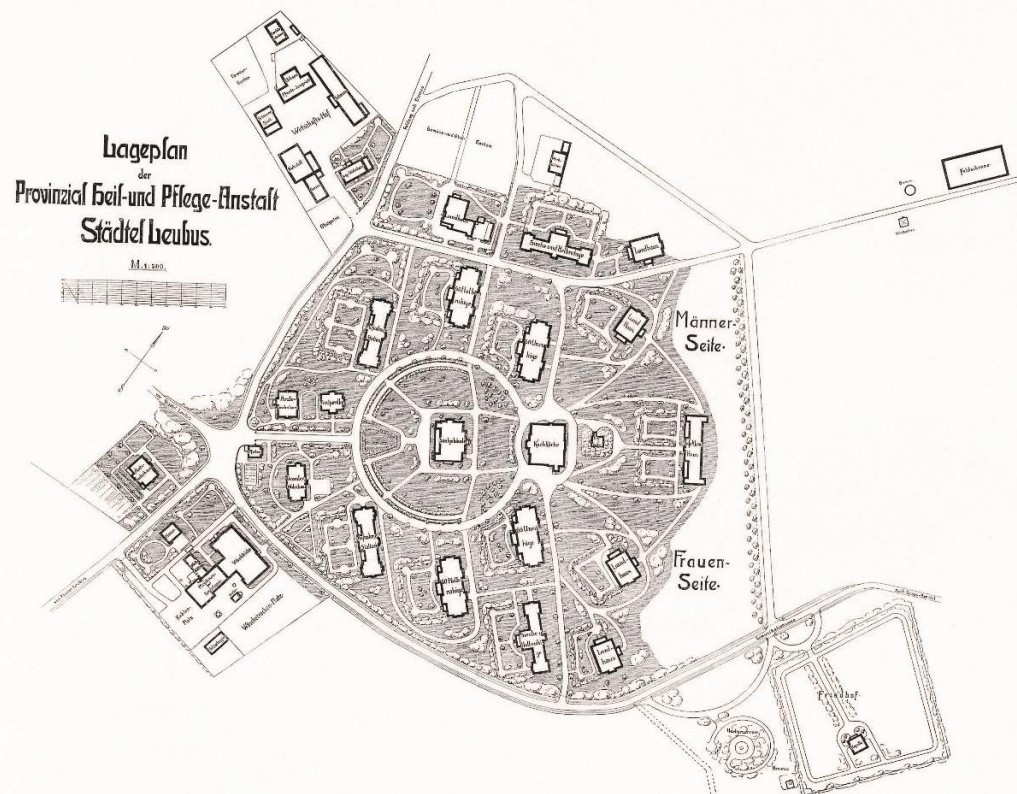

**Figure 4.** A historic plan of the provincial treatment and care facility in the town of Lubiąż; source: [105] public domain.

During World War II, the institution was under Nazi occupation and psychiatric patients of Jewish nationality were partially relocated there and killed during Aktion T4 in unexplained circumstances [106]. From 1942 to 1945, the site was used as a sanatorium for the troops returning from the front. After WWII, it was used for a short time as a field hospital for the Soviet army and later, buildings devastated by war activities were handed over to an organization dealing with agricultural education. The facility was turned into an agricultural mechanization training centre. It was not until 1957 that the institution again became a state-owned psychiatric treatment hospital. Nowadays, it is the most important stationary psychiatric institution for the Lower Silesia region, with seven wards (general and forensic wards and enhanced security wards for adults and adolescents) and a treatment and care facility for the elderly [107].

Nowadays, the time patients spend outside depends on the season and the weather. Most often, it is about an hour a day. There are usually organized group walks accompanied by therapists. During a pandemic, patients are not allowed to leave their unit building unaccompanied. In addition to two general departments, the hospital has two wards with enhanced security (juvenile and adults), two forensic departments, and a long-term care and treatment facility.

Closed wards have their own separate gardens. These areas are separated from each other, and walks are organized according to a fixed schedule. The other gardens are open, without any fences, and not regularly maintained. However, there are only a few benches in the park, and no gazebos or tables. Despite this fact, patients benefit from walks and report that these relax them, calm them down, and improve their mood.

The hospital does not conduct hortitherapy, but organizes artistic therapeutic classes, during which plant elements are used. During the walks, the patients collect flowers and leaves, with which they later make bouquets and paste into compositions on paper. The three-dimensional works are made of cones from the trees that grow in the park. Natural materials work well in occupational therapy because they offer more possibilities and sensory experiences than expensive plastic and stationery materials. Personal creativity provides patients with a great amount of satisfaction, allowing them to focus on something positive and celebrate their achievements when their works are displayed.

The green areas around the pavilions and the layout of the hospital are definitely an advantage, allowing for relaxing walks and enjoyment of the benefits of nature. Despite the transformations and the history of detrimental military use during World War II and the period directly after, a part of the oldest tree stand (plane trees, chestnut trees, and spruces) has survived and is inhabited by many birds. Both patients and staff appreciate it [108].

3.3.3. Weissenhof Weinsberg Klinikum

Klinikum am Weissenhof (nowadays: Zentrum für Psychiatrie Weinsberg) was built from 1900 and opened in 1903 as the fifth state-owned psychiatric institution in the province of Württemberg, initially aimed at providing treatment for 500 patients. To accommodate some separate wards and auxiliary buildings arranged on an organic plan with a circumferential road (Figure 5), the public domain of Weißenhof, a century-old estate, was chosen [105]. The plot was situated in a country area and offered magnificent views of the woody landscape to the southwest, with the romantic Burgruine Weibertreu on the hill. The setting was rural; however, the nearest railway station in Weinsberg was only two kilometres away and the historic city of Heilbronn was just six kilometres afar.

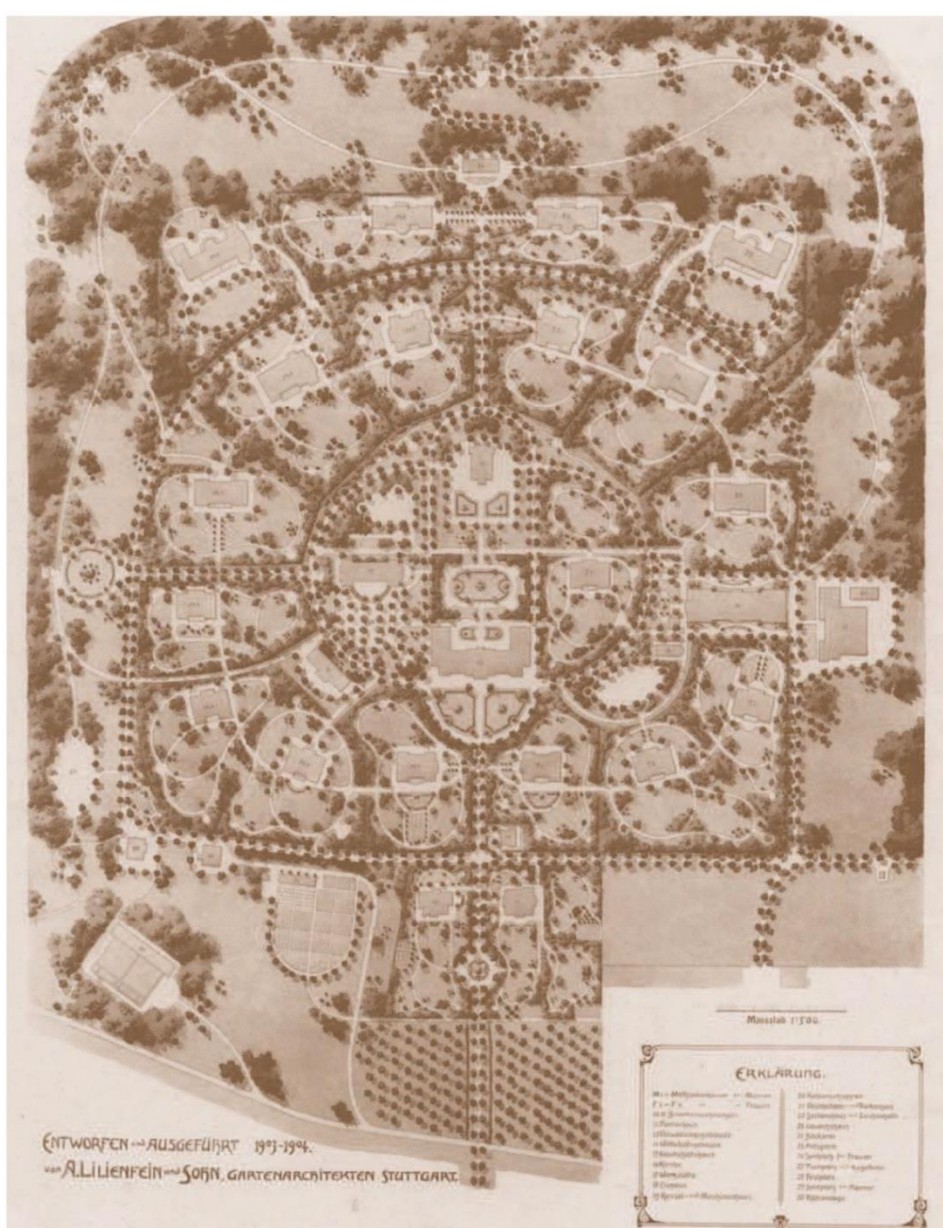

**Figure 5.** A plan of the park and gardens surrounding the institution in Weissenhof-Weinsberg, planned and implemented by A. Lilienfein and son, garden architects in Stuttgart from 1903–1904; source: courtesy of the archive of the Zentrum für Psychiatrie Weinsberg.

The last building of the pavilion system hospital was the institutional church, completed between 1913 and 1915. During the World War II, like many psychiatric hospitals under the national socialist regime in Germany, Aktion T4—a campaign to exterminate psychiatric patients and the mentally ill—took place here. Altogether, around 2000 patients lost their lives, transported to euthanasia centres from Weissenhof. After the war, the institution at Weissenhof-Weinsberg resumed its medical activity and was in part rebuilt. In 2003, a new forensic psychiatry department was built [109].

However, the overall composition remains still clearly visible, including the main longitudinal axis formerly dividing the ensemble into men's and women's sections. Nowadays, the clinic at the Weissenhof also offers treatment for patients other than psychiatric patients, and has 522 stationary beds, which is close to its historic capacity, while only limited cases need long-term stays. Several departments are located in a total of 97 buildings in the 43 ha park. At the time of construction, the park was laid out by a landscape architect, and apart

from the common parts, farm, arable land, a cemetery, and ornamental grounds, each unit had its own garden enclosed by a fence and hedge.

The subjects of pride and care are 3800 trees, some of which are ageing and particularly vulnerable to weather and climate conditions, requiring supplementation, in recent years, by new and resilient species [110].

The historic park is used therapeutically in very different ways with the patients. Many activities within occupational therapy are carried out in the park. On weekdays, the care of the park is taken care of by the patients acting together. These tasks can range from raking leaves to cutting hedges and trees, and are always performed under the guidance and supervision of specialist staff. A small group of patients is also assigned to the animal enclosure within the former hospital farm. On average, 25 patients (divided into several small groups) are present on the site. The patients are usually employed in the park, either in the morning or in the afternoon, for 3 h each time, and always with a break.

A major advantage of occupational therapy in the park is that the result can usually be seen and experienced directly, and this gives patients a direct sense of achievement. In addition, the patients very often receive praise and recognition from passers-by. Since the clinic area has opened, there are many visitors and walkers around, in addition to employees of the clinic, who stop briefly when they pass, and happily let the patients know how beautifully the park has become thanks to the patients' work and involvement.

During the work, the age of the clinic and its park, as well as its history and its change over time, are regularly discussed in conversations. Both the changes in the flora and the park architecture, as well as the political past, are often topics of conversation and arouse the interest of the patients [111]. The dominant feature of the historic park landscape is the division into 'left' and 'right side', because at the time of construction all female patients were treated on one side and all male patients on the other side. It was, therefore, possible to draw a vertical line through the former park as a boundary between women and men, which is not valid anymore, but still can be seen in the overall layout of the site.

There are also many additional therapeutic activities carried out regularly in the park. These include exercise therapy, therapeutic walks (accompanied by professionals), and therapeutic farming. However, it is difficult to assess exactly how many patients are involved altogether in these practices, because several smaller care units which enjoy independence within the institution as a whole operate on the site, and no overall records concerning all therapies related to nature are kept.

## 4. Discussion

According to the American Horticultural Therapy Association (AHTA), horticultural therapy (HT) is a specialized type of therapeutic gardening programme. If one compares actions described in the case studies with the AHTA position paper on horticultural therapy [112], cases from Kobierzyn and Weinsberg are close to the definition. They offer voluntary regular activities focused on gardening tasks with a therapeutic aim. Weissenhof also offers additional programmes in the range of ecotherapy, such as exercise therapy, therapeutic walks (accompanied by professionals), and animal based-interventions within the hospital farm. Kobierzyn also attempts to introduce bee-therapy (apitherapy). In the case of Lubiąż, activities from the spectrum of nature arts and crafts are offered, and staff complains about the insufficient employment of qualified nature therapists.

There is, however, little known about the therapeutic reflection on those particular programmes or a systematic assessment of the well-being of the participants. Moreover, AHTA has its own system of certification and training, which is not a common training scheme for European countries that only, to some extent, have their own associations of horticultural therapists, such as those operating in Germany (Internationale Gesellschaft Gartentherapie e.V.) [113] and Switzerland (Schweizerische Gesellschaft Gartentherapie und Gartenagogik, SGGTA) [114].

One of the main advantages of the use of gardens in historic mental hospitals is that their landscape is perceived in a wider context of the whole institution. It is designed in a

manner such that the whole area resembles an independent settlement, often referred to as a 'garden city', and a world in itself. Therefore, patients can work during their therapeutic gardening sessions in smaller enclosures where the results of their efforts are easier to notice and can be observed daily, which can contribute to their satisfaction by showing their impact on the reality of the garden. Successful shaping of a direct environment can contribute to the improvement of their mood. All the interviewees stressed that the lush greenery of the historic hospital site is an asset of the institution. They also indicated the challenges which result from the age of the trees and the necessary respect for the historic substance of the buildings and its network of paths.

Current research is very much engaged in creating guidelines for designing therapeutic green areas and gardens. Grahn's triangle of supporting environments [115] refers to aspects of both passive and active engagement with nature. This model can be applied in the context of the gardens in historic mental hospitals, indicating that patients with a subjective experience of low well-being are likely to manage inward-directed engagement and take part in more contemplative activities in hospital gardens. This can be a good guideline for the prevalent type of activities in the ward gardens. Consequently, those gardens would be the closest area enabling relaxation and contemplation.

Trojanowska offers a framework to be applied to public park design as an important element of public open spaces (POS) which can promote health and well-being [116]. While all these methodologies are established for the purpose of contemporary landscape and garden design, they can be to some extent useful when assessing therapeutic features of heritage gardens in historic mental hospitals. It is necessary, however, to be aware that those gardens were sometimes created more than 100 years ago and require a balanced approach respecting their cultural value. In addition, they were the product of an outdated understanding of mental disorders and they resulted from certain social circumstances. Nonetheless, they offer a good frame for the whole range of green interventions supporting the mental health and well-being of patients and several sustainable practices to be performed. What is more, some solutions connected with enhancing biodiversity or reducing intensive maintenance of undergrowth to secure bird habitats in historic parks can be problematic from the point of view of patients' subjective security perception. One of the challenges is protecting long-distance vistas, a feature indicated as vital in both historic and contemporary studies. Keeping them clear is one of the most difficult aspects to control, especially in those places where urban sprawl reached mental health hospitals once located on the outskirts of the city, because it does not depend on the institution administration only, but requires sensible urban planning regulations.

An important issue seems to be the need for specialist horticulture skills intended for historic garden maintenance, if it is to be treated as an occupational therapeutical gardening therapy programme. Social enterprises, which employ people suffering from mental disorders, attempt to fill that gap and already operate on some sites of historic mental hospitals, such as in Trieste (La Cooperativa Agricola Monte San Pantaleone [117]) and on the premises of the former Waldhaus Klinik Chur [118]. The latter case is particularly interesting because it offers jobs to adults with mental disabilities and provides them with professional training to become florists and gardeners for ornamental plants within the framework of the sheltered workshop of Graubünden Psychiatric Services (Psychiatrischen Dienste Graubünden-PDGR). The company, apart from other activities, maintains the park and the grounds of the historic Waldhaus Clinic in Chur (Figure 6), which includes cultivating 1159 square meters of flower beds on the premises, mowing, trimming, and scarifying the entire lawn, trimming the hedges, maintaining the flower beds' seasonal flower arrangements, watering, or removing weeds. The company is also responsible for many cleaning tasks in the garden and park, such as emptying the rubbish bins on the site, collecting what has been left behind, cleaning the large water basins, and collecting and disposing of leaves and branches from the trees on the site. The company also offers minor gardening tasks on a contractual basis for private customers from the area [119].

This approach marks an important future research direction, but these cases need further structured analysis that takes into account the long-term impact on patients' lives.

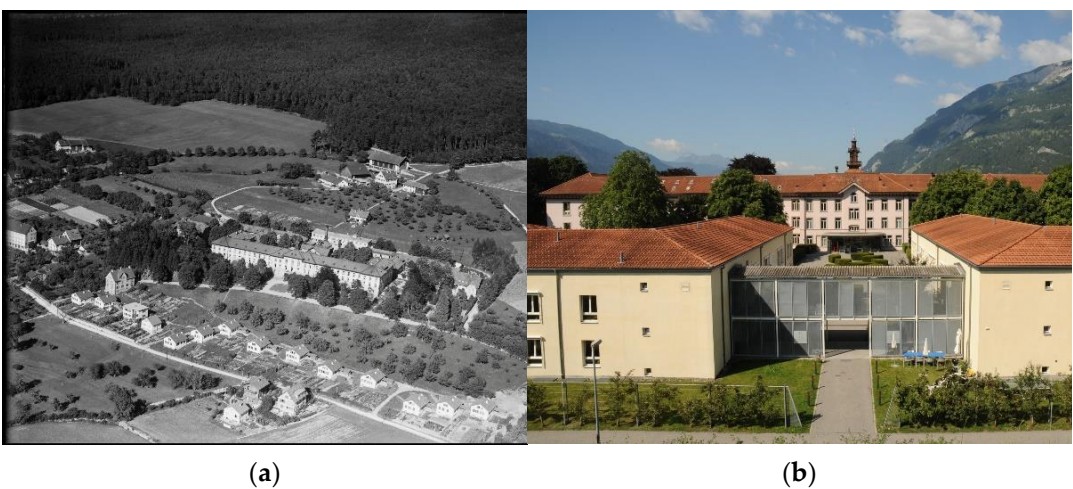

(**a**)                                                       (**b**)

**Figure 6.** (**a**) Chur, Waldhaus Psychiatric Clinic, 1.9.1947, aerial photograph by Werner Friedli; source: ETH-Bibliothek Zürich, [120] CC BY-SA 4.0; (**b**) Contemporary view of Klinik Waldhaus (2011), photo by Cloot, CC BY-SA 4.0; source: [121].

*Ecotherapy Programmes and Their Relation to a Historic Mental Hospital Therapeutic Landscape*

As described above, landscapes and gardens and their use had and still have an important role to play in the construction of the former asylum sites and the well-being of patients. Many connections to nature present in historic mental hospitals still exist, while some of them changed their character because of various reasons, including urban development of the surroundings and significant changes in psychiatric care organisation, to name only two of these reasons.

Table 2 provides a comparison of types of nature connections in historic mental hospitals in the past and nowadays.

**Table 2.** Comparison of the types of connections to nature and gardens in historic mental hospitals in the past and nowadays; source: author's elaboration.

| Connection to Nature in Historic Mental Health Institutions | |
|---|---|
| **In the Past** | **Nowadays** |
| The garden setting of the asylum, located beyond city boundaries, is considered a space of mental escape from illness, and an environment supporting recovery. | Location on the outskirts of cities still considered as pleasant, but often surrounded by residential areas and urban sprawl. |
| Distant views from the institution showing picturesque landscapes. | Views become limited because of contemporary urban development—nevertheless, the most important can be preserved. |
| Gardens as places of physical exercise (gymnastics and walks); daily routine in utility and kitchen gardens; places of occupational therapy; terraces in the sun as therapy for tuberculosis. | Meditation and rest in the gardens; gardening therapy in the ward gardens; social activities—talks and therapeutic group meetings in the open air; therapeutic walks in the parks and woods surrounding the institution. |
| Orchards and arable land for food production as therapeutic work and an economic necessity. | Limited therapeutic farming/agricultural activities and animal tending. |
| Private gardens of the employees of the institution who lived onsite. | Hospital gardens as sites of regeneration during short breaks for employees. |

While contact with nature and activities in the gardens are generally appreciated, there are sometimes concerns about the need for maintenance of the hospital grounds in the aspect of specialist care. The issue of ancient heritage trees and their resistance to climate change-related extreme weather conditions seems particularly demanding [102,111].

Taking this into account, and based on desk research and selected case studies, contemporary ecotherapy programmes can be assigned to particular areas constituting the composition of historic mental hospital gardens today(Table 3). This attribution indicates the potential contemporary use of landscapes for various ecotherapy programmes. While Table 3 presents all possible elements of the typical heritage asylum landscape, implementation of assigned ecotherapy programmes depends on the state of the preservation and integrity of particular sites. In hospital complexes where farming activities were abandoned a long time ago and the arable land was sold, restoring activities aimed at this landscape might be impossible.

**Table 3.** Distinctive elements of the composition of historic mental hospitals, their sites, and their potential therapeutic use for ecotherapy programmes; source: author's elaboration.

| Ecotherapy Programme Opportunities in Historic Mental Hospitals | |
|---|---|
| **Elements of Landscape Composition of Historic Mental Hospital Sites** | **Potential Therapeutic Use for Ecotherapy Programmes** |
| Gardens at the wards. | Social and therapeutic horticulture; nature arts and crafts; green exercise therapy (yoga and open-air gyms). |
| Ornamental grounds and representative entrance areas. | Social and therapeutic horticulture; nature arts and crafts. |
| Landscape park. | Environmental conservation; nature arts and crafts; green exercise therapy (walking). |
| Hospital farm. | Care farming; animal-assisted interventions; animal-assisted therapy. |
| Former kitchen garden (including historic and new glasshouses), orchards, and arable land. | Social and therapeutic horticulture; nature arts and crafts; care farming. |
| Private gardens of the employees of the institution who lived onsite. | Social and therapeutic horticulture; nature arts and crafts. |

On the contrary, gardens adjoining particular ward buildings are easier to restore and offer many opportunities because of their smaller size and direct link to the building where patients spend time. Their composition was less sophisticated than the outline of historic parks surrounding the institution and their maintenance requires mostly basic gardening skills. There is not much historic planting material, apart from a small number of old trees, that require specialist arborist care. The use of annuals, perennials, and herbaceous plants offers an opportunity for engaging all the senses, and creates sensory gardens, which are an acknowledged type of therapeutic garden [122] and can help provide pleasant and meaningful passive restoration.

While the results of the study enable a rough framework for possible ecotherapy programmes' attribution to the distinctive elements of the historic mental hospital landscape, the study has some limitations resulting from the chosen methodology.

It relies on the researcher's interpretation of data provided by interviewees. Here, conceptualizations are based on the lens of an architect with a special research focus and background in historic conservation and the history of gardens. Therefore, further qualitative and quantitative research involving in-depth psychological and medical perspectives

of both patients and therapists is necessary to evaluate the therapeutic effect and clinical significance of ecotherapy conducted in the context of historic mental hospital gardens.

This marks a significant direction for future research, which should be a systematic evaluation of green therapies carried out in institutions located in heritage psychiatric institutions with gardens and parks.

## 5. Conclusions

Therapeutic gardening might be widely propagated as support for other therapies carried out in historic mental hospitals that nowadays house modern mental healthcare facilities. There is an abundance of evidence that therapeutic gardening may improve physical, psychological, and social health, and contact with nature prevents numerous mental health issues facing today's society. However, much still must be done to include green therapies in psychiatric care and encourage the training of professionals to increase people's opportunities and motivation to engage in gardening activities [123].

The main advantage of historic gardens and parks surrounding psychiatric care institutions located on heritage asylum sites is that they can make use of the existing landscape and gardens and parks, instead of waiting for newly planted material to grow around even the most innovative new hospital units built from scratch. This would be benefit historic parks, which are usually listed in monuments' registers as an integral part of the protected heritage buildings of the hospital. In addition, many sustainable gardening practices, which require manual work and specific individual solutions, can be introduced as green therapies. However, this approach, although promising, has some limitations.

Historically, occupational therapy, which included programmes from the spectrum of contemporary green therapies, was usually imposed on patients. In former asylums, patients had to subordinate to doctors' prescriptions concerning their activities and could not decline the imposed work. What is more, patients did not get paid. For this reason, the institution could maintain its often-extensive grounds by relying mostly on their work. This, naturally, does not prevent possible therapeutic outcomes, but today is considered an ambiguous practice and is associated with the exploitation of the patients treated as an available workforce. Nowadays, it is forbidden to employ patients without remuneration. All activities within the therapeutic treatment model are voluntary and take into account patients' independence and preference. Therefore, it would be difficult to rely only on the patients' sometimes irregular or fluctuating participation in garden maintenance to take appropriate care of the historic garden, which requires certain skills and regularity.

However, the example of a social enterprise (or protected workshops) is a particularly promising opportunity for patients recovering from the acute stage of their illness and on their way to independent living. This example offers the possibility of a meaningful occupation and goes beyond the simplest gardening tasks into the direction of specialization, particularly in the maintenance of historic gardens, which require specialist care. Patients working in social enterprises may have financial motivation combined with a sense of gaining independence and self-development.

Taking into account the beneficial outcomes associated with green therapies relying on restoring human connection to nature, and with maintaining the landscape values of historic mental hospital gardens, implementation of ecotherapy may be considered a step toward improving the well-being of the patients and the care for heritage gardens. Ecotherapy also offers an opportunity to involve local communities from the neighbouring housing areas in nature-based interventions on former asylum sites that could benefit from it, as they are often the nearest green space [124]. This could contribute to a sense of place and place attachment to those sometimes difficult and little-known heritage sites. This way, their mental health and well-being can also be improved, by improving their connection to the heritage [125] park and garden of the historic mental hospital.

**Funding:** This research received no external funding.

**Data Availability Statement:** Data available from the author.

**Acknowledgments:** I direct a special thanks to Elżbieta Filek and Maciej Bóbr, (both from Babiński Clinical Hospital in Kraków Kobierzyn), to Agnieszka Kukułka from Lubiąż Psychiatric Hospital, and to Sophia Lager from Zentrum für Psychiatrie Weinsberg, who were my interviewees for the purpose of this study.

**Conflicts of Interest:** The author declares no conflict of interest.

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
