# Peer review of "Gardens of Historic Mental Health Hospitals and Their Potential Use for Green Therapy Purposes"

_land, doi:10.3390/land11101618_

Round 1

Reviewer 1 Report

v  Relevant and current topic that needs to be addressed. Significant literature and data collection.

v  Adequate theoretical frame. A great definition of the concept

v  The implications and expected applications are well explained from the beginning and practical recommendations at the end are clear and realistic.

Author Response

Reviewer 1:

v  Relevant and current topic that needs to be addressed. Significant literature and data collection.

v  Adequate theoretical frame. A great definition of the concept

v  The implications and expected applications are well explained from the beginning and practical recommendations at the end are clear and realistic.

Dear Sir/Madam,

Thank you very much for your appreciation of my research and contribution to the field! The paper was finally revised and the spell check was carried out. 

Reviewer 2 Report

The article approaches an important topic and makes an important contribution to the field. The research has sufficient depth and the manuscript is properly written and well organized, and significant for a broad international audience. I have only two minor suggestions for the introduction. First, I suggest adding to the literature review a paragraph discussing the connection between the historical concept of "gardens" and the modern concept of green infrastructure, especially in relation to seeing therapeutic gardens as a possible ecosystem service (see, for example, http://www.rsdu.ro/Art/RSDUv5a05.pdf, https://doi.org/10.1080/13549839.2011.631993 or https://doi.org/10.1080/02697451003740213); second, the introduction would benefit upon a concluding paragraph stating the research goals boldly, i.e., "this study aims to..."

Author Response

Reviewer 2:

  • The article approaches an important topic and makes an important contribution to the field. The research has sufficient depth and the manuscript is properly written and well organized, and significant for a broad international audience.

Thank you very much for your appreciation of my research and contribution to the field!

  • I have only two minor suggestions for the introduction. First, I suggest adding to the literature review a paragraph discussing the connection between the historical concept of "gardens" and the modern concept of green infrastructure, especially in relation to seeing therapeutic gardens as a possible ecosystem service (see, for example, http://www.rsdu.ro/Art/RSDUv5a05.pdf, https://doi.org/10.1080/13549839.2011.631993 or https://doi.org/10.1080/02697451003740213);

Suggested connection between the historical concept of "gardens" and the modern concept of green infrastructure with appropriate references was added in form of a paragraph in LL: 134-143 

“While gardens have always been an appreciated element [37] of historic cities providing public green leisure areas such as commons, pleasure grounds, and public parks [38], they become nowadays a vital part of green infrastructure (GI) [39] which may provide among other services also therapeutic functions. Historic gardens along with other types of garden spaces [40, 41] may be treated as resources in sustainable urban development in a time of accelerating densification and climate change. Relationships between green infrastructure and ecosystem and human health, construction, evaluation and management of green infrastructure, and analysis of a special aspect of green infrastructure are recently important research topics in the field of landscape planning and urban studies [42]. “

The following papers were added to the list of references:

[37] Saratsi, E.; White, J.; Holyoak, V. Taking account of heritage values of urban parks and gardens, Living With Environmental Change Policy and Practice Notes Note No.36, September 2016, DOI: 10.13140/RG.2.2.13646.25928.

[38] Gilmore, A. The park and the commons: vernacular spaces for everyday participation and cultural value, Cultural Trends2017, 26:1, 34-46, DOI: 10.1080/09548963.2017.1274358.

[39] Mell, I. C. Green infrastructure: reflections on past, present and future praxis, Landscape Research2017, Vol. 42:2, pp.135-145, DOI: 10.1080/01426397.2016.1250875.

[40] Swensen, G.; Krokann Berg, S. The ‘garden city’ in the green infrastructure of the future: learning from the past, Landscape Research2020, Vol. 45:7,  pp. 802-818, DOI: 10.1080/01426397.2020.1798365.

[41] Cameron, T.; Blanuša, J.; Taylor, E.; Salisbury, A.; Halstead, A.J.; Henricot, B.; Thompson, K. The domestic garden – Its contribution to urban green infrastructure, Urban Forestry & Urban Greening, 2012, Volume 11, Issue 2, Pages 129-137, ISSN 1618-8667, https://doi.org/10.1016/j.ufug.2012.01.002.]

[42] Ying, J.; Zhang, X.; Zhang Y.; Bilan S. Green infrastructure: systematic literature review, Economic Research-Ekonomska Istraživanja, 2022, Vol. 35:1, pp.343-366, DOI: 10.1080/1331677X.2021.1893202.

  • second, the introduction would benefit upon a concluding paragraph stating the research goals boldly, i.e., "this study aims to..."

A concluding paragraph was added in LL 178-183:

“Therefore, this study aims to describe relations between landscape and garden composition of heritage mental hospital sites and a spectrum of green therapies in the past and nowadays. Green-based interventions in those particular places will be examined with the reference to historic sources, contemporary literature and research and three selected case studies of contemporary mental health centres operating on historic psychiatric hospital heritage sites. “